# The Association between Social Determinants of Health and Self-Reported Diabetic Retinopathy: An Exploratory Analysis

**DOI:** 10.3390/ijerph18020792

**Published:** 2021-01-18

**Authors:** Emily L. Silverberg, Trevor W. Sterling, Tyler H. Williams, Grettel Castro, Pura Rodriguez de la Vega, Noël C. Barengo

**Affiliations:** 1Department of Translational Medicine, Herbert Wertheim College of Medicine, Florida International University, Miami, FL 33199, USA; esilv074@med.fiu.edu (E.L.S.); tster012@med.fiu.edu (T.W.S.); twill234@med.fiu.edu (T.H.W.); gcastro@fiu.edu (G.C.); rodrigup@fiu.edu (P.R.d.l.V.); 2Department of Public Health, Faculty of Medicine, University of Helsinki, 00100 Helsinki, Finland; 3Department of Epidemiology and Public Health, College of Medicine, Riga Stradins University, LV-1007 Riga, Latvia

**Keywords:** diabetes, diabetic retinopathy, social determinants of health, built environment, economic stability, education, community, race, ethnicity, health care

## Abstract

One-third of Americans with diabetes will develop diabetic retinopathy (DR), the leading cause of blindness in working-age Americans. Social determinants of health (SDOHs) are conditions in a person’s environment that may impact health. The objective of this study was to determine whether there is an association between SDOHs and DR in patients with type II diabetes. This cross-section study used data from the 2018 Behavioral Risk Factor Surveillance System (BRFSS). This study included people with self-reported diabetes in the US in 2018 (n = 60,703). Exposure variables included homeownership, marital status, income, health care coverage, completed level of education, and urban vs. rural environment. The outcome variable was DR. Logistic regression analysis were applied to calculate odds ratios (ORs) and 95% confidence intervals (CIs). Alaskan Native/Native American (OR 2.11; 95% CI: 1.14–3.90), out of work (OR 2.82; 95% CI: 1.62–4.92), unable to work (OR 2.14; 95% CI: 1.57–2.91), did not graduate high school (OR 1.91; 95% CI: 1.30–2.79), only graduated high school (OR 1.43; 95% CI 1.08–1.97), or only attended college or technical school without graduating (OR 1.42; 95% CI: 1.09–1.86) were SDOHs associated with DR in patients with diabetes. Health care providers should identify these possible SDOHs affecting their diabetic patients.

## 1. Introduction

Approximately one in 10 Americans have diabetes while one in three have prediabetes according to the Center for Disease Control and Prevention (CDC) 2020 National Diabetes Statistics Report [1]. Among patients with diabetes mellitus, scientific evidence has revealed that about one third of them will develop diabetic retinopathy (DR) at some point in their lifetime [2]. DR is defined as a neurovascular and microvascular disease in patients with poorly controlled diabetes (i.e., prolonged diabetes with hyperglycemia) that eventually leads to visual impairment [2]. There is a strong and predictable relationship between DR and glycemic control in someone with diabetes mellitus [3,4]. A previous study was performed that examined three different glycemic measures in 44,623 participants aged 20–79 years with gradable retinal photographs. These measures were fasting plasma glucose (FPG), 2 h post-oral plasma glucose (2 h PG), and Hemoglobin A1C. The results of the study determined that an increased prevalence of DR was observed over the ranges of 6.4–6.8 mmol/l for FPG, 9.8–10.6 mmol/l for 2 h PG, and 6.3–6.7% for A1C [4]. Some populations in the United States, such as Native Americans, have an increased incidence of diabetes and an increased incidence of DR. A study conducted in the Pima People showed that 18% of diabetics with a two-hour postprandial glucose reading greater than 200 mg/dL presented with evidence of DR [5]. This comorbidity of DR along with managing one’s diabetes can be debilitating in a person’s life, and this decrease in vision is something that every diabetic patient would like to completely avoid. By taking into account a more comprehensive understanding of each patient beyond his or her clinical symptomatology, health care providers may be able to better treat their patients and help mitigate certain sequelae associated with diabetes mellitus, such as DR.

The overall health of a person is encompassed by more than solely visiting a primary care physician once a year or receiving the annual flu shot. Health involves the ability to access certain social and economic means to facilitate obtaining resources for proper health care. In addition to the tangible differences among those who live in different built environments, social relationships within the community play a large role in the overall health of people. A systematic review conducted in 2014 concluded that social determinants of health (SDOHs) impact glycemic control, which can increase one’s risk for DR [3,6]. Additionally, a study conducted in 2016 found a relationship between food insecurity and poor glycemic control [7]. These relationships have been understated in medicine and are just now being seen as important protective factors for disease. Having a more in-depth understanding of how SDOHs affect health can help providers better treat their patients. Healthy People 2020 lists the following determinants that should be of focus beyond health care: neighborhood and built environment, economic stability, education, and social and community context. Multiple factors or experiences can fall under a specific social determinant. For instance, the social and community context can include civic participation, social cohesion, and racism [8]. Systemic racism can affect multiple SDOHs through residential segregation, as one example [9]. Therefore, understanding race and ethnicity as a social construct can elucidate how SDOHs play a role in health.

Past publications have shown a relationship between certain social determinants of health (SDOHs) and DR. For instance, some studies have revealed an increased risk of DR for those belonging to the minority race and ethnic groups [10,11,12,13], for those of lower economic statuses [10,14], and for those that have less than a high school education [10,14]. Additionally, other studies have shown correlations between a person’s built environment or neighborhood and having DR [15,16].

However, most studies have mainly assessed the mechanism and biochemical processes responsible for the manifestation of this visual impairment in uncontrolled diabetes. There is limited research regarding the role of SDOHs as risk factors for DR. SDOHs are areas of life, besides receiving medical care, that can affect a person’s overall health and well-being. SDOHs include many components of a person’s everyday life, such as education, socioeconomic status, health care, built environment, and connectedness to a community [8,17]. Only a few prior studies have investigated the associations between certain SDOHs and the prevalence of DR. Moreover, the majority of them have been performed outside of the United States and their findings were inconsistent [12,13,14,15,16,18,19,20]. For example, a study conducted in the US in 2018 only contained 29 participants in its sample size, which may not be an adequate number of participants to fully understand how SDOHs are associated with DR [21]. Past literature also shows mixed results on how the built environment affects DR [15,21].

The aim of this study was to assess the association between the SDOHs and the prevalence of DR in the United States population. By determining whether SDOHs are potential risk factors for DR, health care professionals can gain a better understanding of their patients.

## 2. Materials and Methods

### 2.1. Study Design

This study utilized a cross sectional study design and consisted of a secondary analysis of data obtained from the 2018 Behavioral Risk Factor Surveillance System (BRFSS).

### 2.2. Study Population

Data for this study were extracted from the CDC 2018 BRFSS [22]. This system aggregated data collected from a self-reported survey in all 50 states and included Washington DC and three territories in the US. The 2018 survey only included the US territories of Guam and Puerto Rico. However, not all states included every survey question relating to diabetes and its complications. Therefore, the participants included in this study were from the following 17 states that asked the inclusion questions: AL, AZ, DE, DC, GA, IO, ME, MS, NJ, ND, PR, SC, SD, TN, TX, VI, and WI. The information is collected annually via telephone and is administered to noninstitutionalized adults (18 years and older) by the CDC and state health departments. Included are questions on residents’ health-related risk behaviors, chronic health conditions, and use of preventive services. A multistage cluster design is used to produce a nationally representative sample where one adult is randomly selected and interviewed [23]. Specific responses necessary for our study included whether the resident owns or rents a home, their marital and employment status, their health care coverage, education level, and whether they have a cell phone for personal use. Thus, the participants from this study were noninstitutionalized adults (18 years and older) with self-reported diabetes. The main inclusion criteria were as follows: age of 18 years or over; participation in the 2018 BRFSS; and self-reported diagnosis of diabetes. Participants were excluded due to having missing data in any of the variables in the analysis, for DR, demographics, BMI, or SDOHs. Participants were also excluded if they gave one of the following responses to any of the relevant BRFSS items: “Don’t know/Not sure” or “Refused.” A total of 14,810 participants were included in the final analysis.

### 2.3. Study Variables: Social Determinants of Health

The sociodemographic variables in this study included information on whether the participants own or rent a home (community and social context), marital status and race (community and social context), health care coverage (health care), level of income (economic stability), living in an urban or rural setting (built environment), and level of education (education).

Owning or renting a home was assessed with the following item: “Do you own or rent a home?” This variable on the BRFSS is multinomial with the potential responses being “Rent,” “Own,” “Other arrangement.” This variable was made dichotomous with the response of “Rent” categorized as “Yes,” and “Own” or “Other arrangement” categorized as “No.”

Marital status was assessed with the following item: “Are you…?” This variable in the BRFSS is multinomial with potential responses being “Married,” “Divorced,” “Widowed,” “Separated,” “Never married,” or “A member of an unmarried couple.” This variable was made nominal with the categories of “Married or Coupled” including the responses of “Married” and “A member of an unmarried couple,” “Divorced or separated” including responses of “Divorced” and “Separated,” and responses of “Widowed” and “Never married” being their own category.

Health care coverage was evaluated through the question “Do you have any kind of health care coverage, including health insurance, prepaid plans such as HMOs, or government plans such as Medicare, or Indian Health Service?” The collection of data regarding this variable was dichotomous, in that respondents could answer “Yes” or “No” whether they obtain health care.

Employment status, a variable that we determined would assess the economic stability of the participant, was determined by participants answering the question “Are you currently... [regarding employment]” This variable was made nominal with the categories of “Employed and Self Employed,” “Out of Work,” “Retired,” “Unable to Work,” or “Homemaker/Student.”

Living in a rural or urban area, a variable to determine the participants’ built environment, was assessed by the following item: “Urban/rural status.” This variable was dichotomous since the responses on the BRFSS were listed as “Urban counties” or “Rural counties.”

The level of education was assessed by the following item: “What is the highest grade or year of school you completed?” This variable was ordinal. The educational categories were organized into four ordered categories. “Less than a high school degree” included the BRFSS items “Never attended school or only kindergarten,” “Grades 1 through 8 (elementary)” and “Grades 9 through 11 (some high school).” The second category is “High school graduate” which included the BRFSS item “Grade 12 or GED (High school graduate).” The next category is “Some college or technical school” which included the BRFSS item “College 1 year to 3 years (Some college or technical school).” Finally, the fourth category is “College graduate” which included the BRFSS item “College 4 years or more (College graduate).”

The race of the participants was determined through a calculated multiracial race categorization. The categories for race were depicted as “White only,” “Black or African American only,” “American Indian or Alaskan Native only,” “Asian only,” “Native Hawaiian or other Pacific Islander only,” “Other race only,” and “Multiracial.”

Finally, ethnicity was assessed by the following item: “Hispanic, Latino/a, or Spanish origin.” This variable was dichotomous with the responses being either “Yes” or “No.” The “Yes” response included the BRFSS item “Hispanic, Latino/a, or Spanish origin,” while the “No” response included the BRFSS item “Not of Hispanic, Latino/a, or Spanish origin.”

### 2.4. Outcome: Self Reported Diabetic Retinopathy

Having DR was assessed by the following item: “Has a doctor ever told you that diabetes has affected your eyes or that you had retinopathy?” This outcome variable was dichotomous since the responses on the BRFSS were listed as “Yes” or “No.”

### 2.5. Other Variables

Other variables that were included in the study are age, sex, and body mass index (BMI). The age of the participants of the study was displayed utilizing the following ordinal categories presented in the BRFSS questionnaire: Imputed age “18 to 24,” “25 to 29,” “30 to 34,” “35 to 44,” “45 to 54,” “55 to 64,” and “65 or older.” The sex of the participants was assessed by the following item: “What is your sex? or What was your sex at birth? Was it…” The variable was dichotomous with the responses being either “Male” or “Female.” This study classified BMI as follows: BMI < 18.5 kg/m^2^ as underweight, BMI > 18.5 kg/m^2^ and <25.0 kg/m^2^ as normal weight, BMI > 25.0 kg/m^2^ and <30.0 kg/m^2^ as overweight, and BMI > 30.0 kg/m^2^ as obese, according to the World Health Organization.

### 2.6. Statistical Analysis

Data were analyzed with the Stata 15.0 software package (StataCorp. 2017. Stata Statistical Software: Release 15. StataCorp LLC., College Station, TX, USA), and sampling weights were used to adjust for the complex BRFSS sampling design. Data analysis was conducted by first performing a descriptive analysis of the demographics, BMI, and SDOH variables. This was followed by a bivariate analysis via both chi-square tests and *t*-tests to test for categorical (i.e., SDOH) and continuous data (i.e., BMI), respectively, to assess difference in the distribution of variables according to the outcome. Collinearity among independent variables was assessed by calculating pairwise correlations to comply with the assumptions of the independence of logistic regression analysis. Finally, binary logistic regression analysis was used to calculate unadjusted and adjusted odds ratios (ORs) and corresponding 95% confidence intervals (CIs) for the associations between the independent variables with the outcome of DR. The adjusted ORs and CIs were presented in two models, full and reduced. All variables were included in the full model, while only statistically significant and clinically relevant variables were included in the reduced model.

## 3. Results

### 3.1. Baseline Characteristics of the Study Participants

Table 1 outlines the baseline characteristics of the 2018 BRFSS self-reported diabetic participants who answered survey questions in the states or US territories that included the DR questionnaire in the BRFSS telephone survey. A total of 14,810 participants were included in the final analysis. Only 927 individuals, approximately 11%, of the self-reported diabetic participants, were between the ages of 18 and 44 years old. The remaining participants were between the ages of 45 and 64 years old (n = 5454, 43.6%) or 65 years and older (n = 8429, 45.5%). Nearly half of the participants were male (n = 6947, 50.9%) and the other half female (7863, 49.1%). A majority of the participants reported their race as white (n = 10,515, 69.5%). Of the remaining participants, 21.2% identified as black (n = 3035), 2.37% identified as being Alaskan Native or Native American (n = 580), and 6.95% identified as “other” (n = 680). Only 13.3% of the participants reported Hispanic ethnicity (n = 706), while 86.7% reported non-Hispanic ethnicity (n = 14,104). Of the participants, 13.9% of reported a BMI of normal weight or underweight (n = 2116), 30.6% of participants reported being overweight (n = 4556), and 55.6% being obese (n = 8138). Of the participants, 76.3% owned a home (n = 10,955), 19.4% rented their home (n = 3227), and 4.27% participants reported “other” when asked if they owned or rented their home (n = 628). A majority of the participants reported being married or coupled (n = 7159, 55.3%). The rest of the participants were divorced/separated (n = 3008, 18.2%), widowed (n = 2970, 14.6%), or never married (n = 1673, 11.8%). Most of the participants reported active employment, with 31.7% (n = 4112) stating that they were employed or self-employed. The rest of the participants were out of work, (n = 492, 5.24%), retired (n = 7033, 38%), unable to work (n = 2632, 20.5%), or students or homemakers (n = 541, 4.58%). Most of the participants reported having health care coverage (n = 13,985, 90.9%), while the remaining participants did not have health care coverage (n = 825, 9.06%). Nearly a quarter of the participants did not complete high school (n = 1780, 22.2%) while the rest of the individuals completed high school or continued beyond high school. Of the participants, 30.9% graduated from high school (n = 4837), 30.2% of participants attended college or technical school but did not graduate (n = 4279), and 16.7% graduated college or technical school (n = 3914). Finally, a majority of participants reported living in an urban area (n = 11,487, 86.9%) while the rest of the participants lived in a rural setting (n = 3323, 13.1%).

### 3.2. Unadjusted and Adjusted Associations of Social Determinants of Health and Diabetic Retinopathy

Table 2 reveals the unadjusted and adjusted associations of SDOHs and DR. Those aged 45–64 had a 50% increase in the odds ratio (OR 1.50; 95% CI: 1.02–2.21) compared with patients 18–44 years of age. Alaskan Native/Native Americans had an increased chance of having DR (OR 2.15; 95% CI: 1.16–3.98) compared with whites. Hispanics had a 61% increase in the odds ratio (OR 1.61; 95% CI: 1.02–2.53) compared with non-Hispanics. Those out of work had a statistically significant 193% increase in the odds ratio (OR 2.93; 95% CI: 1.67–5.13), those unable to work had a 112% increase in the odds ratio (OR 2.12; 95% CI: 1.57–2.89), and those who reported being a homemaker/student had a 55% decrease in the odds ratio (OR 0.45; 95% CI: 0.26–0.79) compared with those who were employed/self-employed. People who did not graduate school had an 86% increased odds ratio (OR 1.86; 95% CI: 1.26–2.73) of DR compared with those who graduated college or technical school, while those who graduated high school or attended college or technical school had a 40% increased likelihood (OR 1.40; 95% CI: 1.07–1.83). However, 65+ years of age, sex, black race and those reporting as “other,” BMI, housing status, marital status, people who were retired, and living place were not associated with DR. Table 2 is the reduced model. People aged 45–64 had a 52% increased chance of having DR (OR 1.52; 95% CI: 1.05–2.21) compared with people aged 18–44. Alaskan Native/Native Americans had a 111% increase in the odds ratio (OR 2.11; 95% CI: 1.14–3.90) compared with whites. Compared to people reporting as employed/self-employed, those out of work had a 182% increase in the odds ratio (OR 2.82; 95% CI: 1.62–4.92), those unable to work had a 114% increase (OR 2.14; 95% CI: 1.57–2.91), and those reporting as a homemaker/student had a 53% decrease (OR 0.47; 95% CI: 0.27–0.80). Compared to people who graduated from college or technical school, those who did not graduate from high school had a 91% increased chance of having DR (OR 1.91; 95% CI: 1.30–2.79), those who graduated high school had a 43% increased risk (OR 1.43; 95% CI: 1.08–1.97), and those who attended college or technical school had a 42% increased chance (OR 1.42; 95% CI: 1.09–1.86). Being 65+ years of age, sex, black race and those reporting as “other,” Hispanic, and those who were retired were not associated with DR.

## 4. Discussion

Our data revealed that the factors of between the ages of 45 and 64, who are Alaskan Native/Native American, out of work, unable to work, did not graduate high school, graduated high school, or attended college or technical school without graduating were associated with DR in patients with diabetes. However, BMI, home ownership, marital status, belonging to an urban or rural demographic, those 65+ years of age, sex, black race and those reporting as “other,” Hispanic, and those who were retired were not associated with DR in our study population. Of important note, Alaskan Native/Native Americans had the strongest association of having DR compared to other races and compared to most other SDOH categories.

Past publications have reported SDOHs and their effects on DR. In general, being in the minority race and ethnic groups increases your risk of having DR [10,11,12,13]. Of the studies conducted in the United States, there was a higher prevalence of self-reported DR in blacks or non-Hispanic blacks compared to whites or non-Hispanic whites [10,11]. Our data contradict this finding by showing no statistically significant association of DR in self-reported black participants (OR 1.18; 95% CI: 0.93–1.49). A study published in 2019, sampling 16,976 US adults with diabetes in North Carolina, reported an increased prevalence of DR in the Hispanic population [11]. In contrast, our data show no association between Hispanic ethnicity and DR. This may be related to low power or other unknown confounders. However, our data did show that those who reported being Alaskan Native or Native American were twice as likely to have DR compared with white participants (OR 2.11; 95% CI: 1.14–3.90). Studies exploring the relationship between health care and DR are limited. One study conducted in Brazil found that patients with type I diabetes had lower rates of diabetes-related chronic conditions, mainly retinopathy, when having private health care compared to public health care through the Brazilian national health care system (OR 1.40, *P* = 0.013) [18]. This study did not provide much insight on how health care coverage affects diabetics in the United States. Our study was unable to show significant increased odds of having DR when lacking health care coverage. Health care coverage should be an important SDOH, but few publications have illustrated the relationship between health care and DR. Further, just because someone has health care coverage, it does not ensure the care they receive is adequate. As an example, it has been shown that Alaskan Native/Native American (AN/NA) people are around 3.2 times more likely than all US races to develop diabetes mellitus. Health care experts believe that this staggering proportion of diabetes in this population may be due to inadequate funding of the Indian Health Service (IHS) [24].

Multiple studies have shown strong relationships between economic status and DR. In general, past research has shown that a low socioeconomic status (i.e., low income, receiving public assistance, or inconsistent access to employment) presents a higher risk of having DR [10,14]. A cross sectional study conducted in the United States in 2012 used the National Health and Nutrition Examination Survey (NHANES) III and NHANES 2005–2008 surveys to analyze the prevalence of having DR in a total of 13,912 diabetic patients [10]. According to the NHANES 2005–2008 surveys, the prevalence of DR was significantly higher among those below the federal poverty level (5.1%; 95% CI 3.4–7.8) compared to individuals above the federal poverty level, especially those four times or greater above the threshold (2.2%; 95% CI: 1.4–3.5) [10]. A Japanese cross sectional study published in 2017 surveyed 672 type II diabetic patients with DR and found an increased odds of having DR in patients with irregular employment or no employment [14]. Our research aligned with these past findings, demonstrating an association between employment status or income and having DR. Additionally, education level is an SDOH that can be related to socioeconomic status and DR. Current scientific evidence has shown that the prevalence of DR is significantly higher in those that have less than a high school education than those with higher levels of education [10,14]. According to the NHANES 2005–2008 surveys, the prevalence of DR was significantly higher among those with less than a high school education (6.9%; 95% CI 5.1–9.2) than among those with more than a high school education (2.7%; 95% CI: 2.1–3.6) [10]. A Japanese cross sectional study published in 2017 that surveyed 672 type II diabetic patients with DR also found that the odds of having DR were higher among junior high school graduates compared to those with more schooling [14]. Our study also found that a higher level of education equated to a lower risk of having DR. A 2016 study conducted in rural parts of India in 105 diabetic patients demonstrated a lack of knowledge of diabetic eye complications in the population, providing a possible explanation for the relationship between lower education levels and having DR [19].

Neighborhood and built environment is an SDOH that needs more research to explore the relationship between this determinant of health and DR. Two studies conducted in Europe explored the relationships between deprived areas, based on the 2007 Index of Multiple Deprivation (IMD) score and the Scottish Index of Multiple Deprivation (SIMD) 2012 score, and DR [15,16]. One study found that those living in deprived areas were less likely to make a screening appointment for DR and the other study found an association between increased SIMD scores and the prevalence of DR [15,16]. In order to evaluate neighborhood and built environment, our study explored participants’ social relationships, home ownership, and urban or rural status. We found no statistically significant association between relationship status, homeownership, or rural and urban status and having DR. Neighborhood and built environment is a complicated SDOH that is difficult to measure and study but is an important factor that has the potential to affect one’s health. For instance, assuming that urban or rural status conveys all the resources available within one’s built environment may introduce some additional bias. Contrary to previous thought, it has been shown that impoverished areas within urban environments can be more deficient in resources than rural areas [25]. These subtleties were not addressed through the survey question on “urban or rural status.”

Research has shown that diabetic patients who delay receiving eye care or obtaining DR screenings in a timely fashion are at an increased risk of developing DR [20]. Specifically, it has been shown that AN/NA populations have a decreased prevalence of DR by as much as 50% from earlier studies due to an Indian Health Service–Joslin Vision Network Teleophthalmology Program (IHS-JVN) that increased their access to care [26]. However, the biochemical processes that specifically lead to a diabetic developing DR are very novel and the etiology is quite unknown. Some of the proposed mechanisms have relied on metabolomics, or the study of unregulated metabolites in biological organisms, to elucidate which disturbed pathways seen in diabetic patients are truly correlated with the progression of DR. These metabolomic studies utilized vitreous samples obtained from diabetics undergoing ocular surgery and identified metabolite markers unique to DR. Current research has proposed that alterations in the pentose phosphate pathway, arginine to proline pathway, or the ascorbic acid pathway seen in diabetic patients can progress to the dysregulation of the neurovascular connection to the retina, eventually resulting in DR [27]. Due to the abnormal increase in metabolites in unregulated diabetes and diabetic patients who do not receive prompt DR screenings, there will be sequelae of retinal inflammation and non-reversible damage to the eye [1].

There are some limitations in our study. Since the data were from the BRFSS survey, and information is self-reported, and our data are subject to information bias. Moreover, the diagnosis of diabetes and DR was self-reported, which underestimates the true prevalence of both diabetes and DR. However, it has been shown that the self-reporting of medical diagnoses has a rather close concordance with medical records [28]. While there was also no distinction made between diabetes mellitus type I and type II, the CDC reports that 90–95% of all patients with diabetes have type II [1]. Hence, it can be assumed that the majority of respondents in the study who had self-reported diabetes had type II. Another limitation of this study is missing data. There were data missing for 3547 people out of 18,357. Less than 10% of participants were missing from each category, but missing data may still play a role in limiting this study. Delay in receiving screenings for DR can increase the risk of having that condition. However, this was not reported by respondents and may be a possible source of confounding. Since DR is a multifactorial complication of diabetes mellitus and is still not well understood, there could also be other sources of confounding not accounted for. For instance, genetic factors may play a role in this complication, but this has not been confirmed. Additionally, there may be other SDOHs and environmental factors affecting the outcome of DR that were not included in this study, such as food insecurity, civic engagement, and health literacy. Finally, since this is a cross sectional study, causality cannot be inferred.

## 5. Conclusions

In conclusion, our data showed that some SDOHs, such as education, economic stability, and race in a social context, can be associated with the outcome of DR in diabetic patients. Thus, it is recommended to identify these determinants when treating patients with diabetes mellitus since delayed screenings can increase the risk of developing DR. While this study has identified SDOHs that may contribute to the outcome of having DR, future studies should be conducted to figure out why this is. Obtaining a better understanding of the mechanisms behind the SDOHs recognized as having an association with DR could aid greatly in a more comprehensive treatment plan for diabetic patients. More research is necessary to determine other possible factors at play not observed in this study. Furthermore, future research should study type I and type II diabetes separately since there are differences in disease pathophysiology and in the disease populations (i.e., insulin use and BMI). This study can also be referenced to better understand the exact relationship between more elusive SDOHs, such as built environment and social relationships. Since the exact pathophysiology of DR is still unknown, health care providers must aim to prevent the onset of disease in which SDOHs must be taken into account.

## Figures and Tables

**Table 1 ijerph-18-00792-t001:** Baseline characteristics of the study participants.

Characteristics	N *	% *
**Age (years)**		
18–44	927	11
45–64	5454	43.6
65+	8429	45.4
**Sex**		
Male	6947	50.9
Female	7863	49.1
**Race**		
White	10,515	69.5
Black	3035	21.2
AN/NA **	580	2.37
Other	680	6.95
**Ethnicity**		
Hispanic	706	13.3
Non-Hispanic	14,104	86.7
**BMI**		
Underweight/Normal Weight	2116	13.9
Overweight	4556	30.6
Obese	8138	55.6
**Home Ownership**		
Own	10,955	76.3
Rent	3227	19.4
Other	628	4.27
**Marital Status**		
Married/Coupled	7159	55.3
Divorced/Separated	3008	18.2
Widowed	2970	14.6
Never Married	1673	11.8
**Employment Status**		
Employed and Self Employed	4112	31.7
Out of work	492	5.24
Retired	7033	38
Unable to work	2632	20.5
Homemaker/Student	541	4.58
**Health Care Coverage**		
Yes	13,985	90.9
No	825	9.06
**Education Levels**		
Did Not Complete High School	1780	22.2
Graduated High School	4837	30.9
Attended College or Technical School	4279	30.2
Graduated College or Technical School	3914	16.7
**Urban or Rural**		
Urban	11,487	86.9
Rural	3323	13.1

* This table includes unweighted counts and weighted percentages; ** Alaskan Native/Native American.

**Table 2 ijerph-18-00792-t002:** Unadjusted and adjusted associations of social determinants of health and diabetic retinopathy.

Characteristics	Unadjusted	Adjusted	
		Full Model	Reduced Model
	OR ^1^ (95% CI ^2^)	OR (95% CI)	OR (95% CI)
**Age (years)**			
18–44	Ref ^3^	Ref	Ref
45–64	1.52 (1.07–2.17)	1.50 (1.02–2.21)	1.52 (1.05–2.21)
65+	1.14 (0.80–1.63)	1.38 (0.89–2.14)	1.45 (0.94–2.33)
**Sex**			
Male	Ref	Ref	Ref
Female	0.84 (0.67–1.06)	0.80 (0.64–1.00)	0.80 (0.64–1.00)
**Race**			
White	Ref	Ref	Ref
Black	1.25 (0.99–1.59)	1.21 (0.96–1.53)	1.18 (0.93–1.49)
AN/NA	2.39 (1.16–4.93)	2.15 (1.16–3.98)	2.11 (1.14–3.90)
Other	1.39 (0.91–2.11)	1.11 (0.64–1.90)	1.11 (0.64–1.93)
**Ethnicity**			
Non-Hispanic	Ref	Ref	Ref
Hispanic	1.74 (1.15–2.65)	1.61 (1.02–2.53)	1.56 (0.99–2.46)
**BMI**			
Underweight/Normal Weight	Ref	Ref	
Overweight	1.12 (0.79–1.59)	1.13 (0.80–1.58)	
Obese	1.01 (0.74–0.1.37)	0.94 (0.69–1.29)	
**Home Ownership**			
Own	Ref	Ref	
Rent	1.37 (1.06–1.76)	1.16 (0.91–1.48)	
Other	1.53 (1.04–2.24)	1.27 (0.85–1.89)	
**Marital Status**			
Married/Coupled	Ref	Ref	
Divorced/Separated	1.14 (0.88–1.49)	0.87 (0.68–1.12)	
Widowed	0.99 (0.71–1.37)	0.97 (0.69–1.35)	
Never Married	1.05 (0.72–1.51)	0.80 (0.57–1.12)	
**Employment Status**			
Employed/Self-employed	Ref	Ref	Ref
Out of Work	3.04 (1.69–5.47)	2.93 (1.67–5.13)	2.82 (1.62–4.92)
Retired	1.04 (0.80–1.34)	1.02 (0.73–1.43)	1.03 (0.74–1.44)
Unable to Work	2.53 (1.86–3.44)	2.12 (1.57–2.89)	2.14 (1.57–2.91)
Homemaker/Student	0.53 (0.34–0.84)	0.45 (0.26–0.79)	0.47 (0.27–0.80)
**Health Care Coverage**			
Yes	Ref	Ref	
No	1.12 (0.75–1.69)	0.90 (0.59–1.38)	
**Education Levels**			
Graduated College or Technical School	Ref	Ref	Ref
Did Not Graduate High School	2.77 (1.95–3.93)	1.86 (1.26–2.73)	1.91 (1.30–2.79)
Graduated High School	1.66 (1.28–2.14)	1.40 (1.07–1.83)	1.43 (1.08–1.97)
Attended College or Technical School	1.56 (1.21–2.00)	1.40 (1.07–1.83)	1.42 (1.09–1.86)
**Urban or Rural**			
Urban	Ref	Ref	
Rural	1.23 (0.91–1.67)	1.27 (0.94–1.72)	

^1^ Odds Ratio, ^2^ Confidence Interval, ^3^ Reference

## Data Availability

Publicly available datasets were analyzed in this study. This data can be found here: https://www.cdc.gov/brfss/annual_data/annual_2019.html.

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
