# Peer review of "The Association between Social Determinants of Health and Self-Reported Diabetic Retinopathy: An Exploratory Analysis"

_ijerph, 2021, doi:10.3390/ijerph18020792_

Round 1

Reviewer 1 Report

An in-depth discussion of SDOH is warranted in the introduction given the scope of this manuscript. Provide the reader with background on which SDOH are important to DM and DR specifically. While race/ethnicity is a social construct, you will need to explain how/why you are conceptualizing it as a SDOH; is it a proxy for something else?

Add sample N to section 2.2

Line 74 provide a citation for CDC 2018 BRFSS

Line 93 correct "Variabels"

Remove ethnicity from 2.5 as you already claimed it in 2.3

Provide citation for "Stata 15.0" line 150

Please consider using subheaders to break up section 3.1. It’s difficult to read as currently formatted.

Line 186, please remove "only" from “only graduated from HS.” As you know for some people opportunities for education are limited. Same with lines 220-222

Line 225, “were not predictors” since these are cross-sectional data, you can’t claim predictability. Please revise.

In general, the introduction to this manuscript require more analysis of SDOH. Set the reader up to understand the influence of SDOH on diabetes and the understanding that the risk of Diabetic Retinopathy increased with DM risk.

Also, the American Indian population has a higher risk, prevalence, and incidence of type 2 diabetes so its also most expected that their odds of DR are increased. Furthermore, the insurance question does ask about adequacy of care. Indian Health Service is underfunded, AI/AN individuals have access to care but it might not be adequate care.

A simple google search of “Diabetic Retinopathy, American Indian” resulted in articles that could help inform the discussion and introduction.

The variable of “Urban/rural status” as a proxy to quality of built environment is weak and this weakness needs to be addressed. I understand the authors are limited to the variables collected in the study but a discussion is warranted.

Line 304-05, “Since DR is a multifactorial complication of diabetes mellitus and is still not well understood, there could also be other sources of confounding not accounted for,” please expand on this statement.

Please consider adding more research evidence from US context as the sample is US based. The studies from Brazil and Japan, for example, are interesting but the context is very different.  

Author Response

Thank you very much for all your valuable comments and guidance to improve our manuscript. We have carefully studied your comments and revised our work according to your suggestions. Your feedback helps us to become better researchers and are highly appreciated.

Comment#1: An in-depth discussion of SDOH is warranted in the introduction given the scope of this manuscript. Provide the reader with background on which SDOH are important to DM and DR specifically. While race/ethnicity is a social construct, you will need to explain how/why you are conceptualizing it as a SDOH; is it a proxy for something else?

Response#1: Thank you for this comment. We agree with the reviewer. We have added the paragraph below to the introduction in the manuscript (lines 62-77) to provide a better in-depth discussion of SDOH in the introduction:

The overall health of a person is encompassed by more than solely visiting a primary care physician once a year or receiving the annual flu shot. Health involves the ability to access certain social and economic means to facilitate obtaining resources for proper healthcare. In addition to the tangible differences among those who live in different built environments, social relationships within the community play a large role in the overall health of people. A systematic review conducted in 2014 concluded that SDOH impact glycemic control, which can increase one’s risk for DR [3,6]. Additionally, a study conducted in 2016 found a relationship between food insecurity and poor glycemic control [7]. These relationships have been understated in medicine and are just now being seen as important protective factors for disease. Having a more in-depth understanding of how SDOH affect health can help providers better treat their patients. Healthy People 2020 lists the following determinants that should be of focus beyond healthcare: neighborhood and built environment, economic stability, education, and social and community context. Multiple factors or experiences can fall under a specific social determinant. For instance, social and community context can include civic participation, social cohesion, and racism [8]. Systemic racism can affect multiple SDOH through residential segregation, as one example [9]. Therefore, understanding race and ethnicity as a social construct can elucidate how SDOH play a role in health.  

Comment#2: Add sample N to section 2.2.

Response#2: We added the sample N of our study to the first paragraph of section 2.2, or line 120

Comment#3: Line 74 provide a citation for CDC 2018 BRFSS

Response#3: We have added a citation for the CDC 2018 BRFSS data and documentation (line 102).

Comment#4: Line 93 correct "Variabls"

Response#4: Thank you for pointing this out. The spelling has been corrected on line 121.

Comment#5: Remove ethnicity from 2.5 as you already claimed it in 2.3

Response#5: Thank you for this comment, we deleted the word “ethnicity” in line 169, section 2.5.

Comment#6: Provide citation for "Stata 15.0" line 150

Response#6: We have added a citation for Stata 15.0 in line 178-179.

Comment#7: Please consider using subheaders to break up section 3.1. It’s difficult to read as currently formatted.

Response#7: Thank you for this suggestion. We added a subheading to properly address the second half of the results section. You will find this addition on line 225.

Comment#8: Line 186, please remove "only" from “only graduated from HS.” As you know for some people opportunities for education are limited. Same with lines 220-222

Response#8: Thank you for this comment. We originally included the word “only” to specify the participant’s last level of education reached. However, we agree that this language could be disparaging, and we have removed the word “only.” (Lines 215-216 & 246.)

Comment#9: Line 225, “were not predictors” since these are cross-sectional data, you can’t claim predictability. Please revise.

Response#9: We replaced “were not predictors” with “were not associated with DR,” line 255-256.

Comment#10: In general, the introduction to this manuscript require more analysis of SDOH. Set the reader up to understand the influence of SDOH on diabetes and the understanding that the risk of Diabetic Retinopathy increased with DM risk.

Response#10: We have addressed the influence of SDOH on diabetes in comment 1. We have also added lines 47-53 to describe the increased risk of diabetic retinopathy with the increased risk of DM.

There is a strong and predictable relationship between DR and glycemic control in someone with diabetes mellitus [3,4]. A previous study was performed that examined three different glycemic measures in 44,623 participants aged 20-79 years with gradable retinal photographs. These measures were fasting plasma glucose (FPG), 2-hour post oral plasma glucose (2-h PG), and A1C. The results of the study determined that an increased prevalence of DR was observed over the ranges of 6.4-6.8 mmol/l for FPG, 9.8-10.6 mmol/l for 2-h PG, and 6.3-6.7% for A1C [4].

Comment#11: Also, the American Indian population has a higher risk, prevalence, and incidence of type 2 diabetes so its also most expected that their odds of DR are increased. Furthermore, the insurance question does ask about adequacy of care. Indian Health Service is underfunded, AI/AN individuals have access to care but it might not be adequate care.

A simple google search of “Diabetic Retinopathy, American Indian” resulted in articles that could help inform the discussion and introduction.

Response#11: Thank you for this comment. We have added the following lines to the introduction and discussion section:

Lines 53-56: Some populations in the United States, such as Native Americans, have an increased incidence of diabetes and an increased incidence of DR. A study conducted in the Pima People showed that 18% of diabetics with a two-hour postprandial glucose reading greater than 200 mg/dL presented with evidence of DR [5].

Lines 276-280: Further, just because someone has health care coverage, it does not ensure the care they receive is adequate.  As an example, it has been shown that AN/NA people are around 3.2 times more likely than all U.S. races to develop diabetes mellitus. Healthcare experts believe that this staggering proportion of diabetes in this population may be due to inadequate funding of the Indian Health Service (IHS).

Lines 320-323: Specifically, it has been shown that AN/NA populations have decreased prevalence of DR by as much as 50% from earlier studies due to an Indian Health Service-Joslin Vision Network Teleophthalmology Program (IHS-JVN) that increased their access to care [26].

Comment#12: The variable of “Urban/rural status” as a proxy to quality of built environment is weak and this weakness needs to be addressed. I understand the authors are limited to the variables collected in the study but a discussion is warranted.

Response#12: We have added the following sentences to the manuscript to address this comment (Lines 314-318):

For instance, assuming that urban or rural status conveys all the resources available within one’s built environment may introduce some additional bias. Contrary to previous thought, it has been shown that impoverished areas within urban environments can be more deficient in resources than rural areas [25]. These subtleties were not addressed through the survey question: “urban or rural status.”

Comment#13: Line 304-05, “Since DR is a multifactorial complication of diabetes mellitus and is still not well understood, there could also be other sources of confounding not accounted for,” please expand on this statement.

Response#13: We have added the following lines to expand on this statement:

For instance, genetic factors may play a role in this complication, but this has not been confirmed. Additionally, there may be other SDOH and environmental factors affecting the outcome of DR that were not included in this study, such as food insecurity, civic engagement, and health literacy (346-349).

Comment#14: Please consider adding more research evidence from US context as the sample is US based. The studies from Brazil and Japan, for example, are interesting but the context is very different.  

Response#14: Thank you for this comment. During our literature review we found only a limited number of studies on this topic conducted in the United States.  We specifically addressed this issue in lines 89-90 to bring greater context to the purpose of our study. We added lines (90-93) to further clarify this issue.

Lines 90-93: For example, a study conducted in the US in 2018 only contained 29 participants in its sample size, which may not be an adequate number of participants to fully understand how SDOHs are associated with DR [16].

Reviewer 2 Report

In the current manuscript, the association between the SDOH and the prevalence of DR in the United States population was assessed. This study is markedly innovative and important for the related research field. Authors showed a sufficient background, and the research design and methods adopted by authors were appropriate. Moreover, the conclusion could be well supported by the results. Some minor issues are needed to be figured out before the manuscript is accepted.

  1. Please add more keywords in order to let more readers to search your study.
  2. Please add a subtitle to describe the data about Table 2.
  3. Please simplify the section of the conclusion. Also, the reference should not occur in this part.

Author Response

Thank you very much for all your valuable comments and guidance to improve our manuscript. We have carefully studied your comments and revised our work according to your suggestions.

Comment#1: Please add more keywords in order to let more readers to search your study.

Response#1: Thank you for this suggestion. We have now added more keywords (see lines 38-39).

Comment#2: Please add a subtitle to describe the data about Table 2.

Response#2: Thank you for this comment, we have added section 3.2 to line 225.

Comment#3: Please simplify the section of the conclusion. Also, the reference should not occur in this part.

Response#3: Thank you for pointing out the reference, we have deleted it. We have also revised the conclusion paragraph. Lines 352-365. The conclusion reads now as follows:

In conclusion, our data showed that some SDOH, such as education, economic stability, and race in social context, can be associated with the outcome of DR in diabetic patients. Thus, it is recommended to identify these determinants when treating patients with diabetes mellitus since delayed screenings can increase the risk of developing DR. While this study has identified SDOH that may contribute to the outcome of having DR, future studies should be conducted to figure out why this is. Obtaining a better understanding of the mechanisms behind the SDOH recognized as having an association with DR could aid greatly in a more comprehensive treatment plan for diabetic patients. More research is necessary to determine other possible factors at play not observed in this study. Furthermore, future research should study type I and type II diabetes separately since there are differences in disease pathophysiology and in the disease populations (i.e. insulin use and BMI). This study can also be referenced to better understand the exact relationship between more elusive SDOH, such as built environment and social relationships. Since the exact pathophysiology of DR is still unknown, health care providers must aim to prevent the onset of disease in which SDOH must be taken into account.

Reviewer 3 Report

I have read with interest this paper which investigates the associations between social determinants and diabetic retinopathy. This is a debated topic that deserves attention as the diffusion of diabetes is constantly increasing and retinopathy is a common complication that often leads to blindness.

Precisely for the interest of the subject I believe that the paper can be improved and therefore I suggest some ideas for its revision.

MAJOR REVISIONS

Title

“…association between social determinants of health and diabetic retinopathy…”. This can be somehow misleading. As the Authors explain (lines 137-138) “Having DR was assessed by the following item: “Has a doctor ever told you that diabetes has affected your eyes or that you had retinopathy?”. Therefore, the association appears in fact to be between social determinants and awareness of having diabetic retinopathy. I recommend that the title should be changed.

Lines 128-135

This paragraph is a bit confusing. Does it mean that each racial group was divided into Hispanic and non-Hispanic according to ethnicity?

Statistical analysis

Correctly, the authors have first obtained a descriptive analysis of their population. Then they have used logistic regression due to the fact diabetic retinopathy is dichotomous (yes or no). I wonder, however, if a different approach could have been more satisfactory. For example, the authors could have tried to find clusters of conditions that are more frequently associated with diabetic retinopathy. In this case multiple component analysis, using diabetic retinopathy as a passive variable could have been more satisfactory. The authors should explain their choice more in detail.

Line 183

“Most of the participants reported having health care coverage (n=13,985, 90.9%)”. On the basis of this observation it appears more reasonable not to consider this as a variable since almost all of the population of the study had health coverage. It could be interesting to know which factors are associated with DR in the small proportion of patient without a health insurance. Moreover “…a majority of participants reported living in an urban area (n=11,487, 86.9%)” (line 189). Are these the same who also have health coverage? If it is so it would be interesting to compare them with participants living in rural areas and without health coverage.

MINOR CHANGES

Line 65

“…on how built environment effects DR…” Do the authors really mean that built environment produces DR, or may be they are saying that it is one of the factors that affects DR?

Line 195

“…Those ages 45-64” maybe should be typed “…Those aged 45-64”.

A 3.2 heading should be placed between line 194 and line 195.

Author Response

Thank you very much for all your valuable comments and guidance to improve our manuscript. We have carefully studied your comments and revised our work according to your suggestions.

Comment#1: Title: “…association between social determinants of health and diabetic retinopathy…”. This can be somehow misleading. As the Authors explain (lines 137-138) “Having DR was assessed by the following item: “Has a doctor ever told you that diabetes has affected your eyes or that you had retinopathy?”. Therefore, the association appears in fact to be between social determinants and awareness of having diabetic retinopathy. I recommend that the title should be changed.

Response#1: Thank you for this comment. The title can be misleading since the survey participants are essentially self-reporting that they have diabetic retinopathy and we cannot guarantee that every participant was diagnosed with diabetic retinopathy. We have adjusted the title to “Self-Reported Diabetic Retinopathy.” We hope you accept the new, revised title.

Comment#2: Lines 128-135: This paragraph is a bit confusing. Does it mean that each racial group was divided into Hispanic and non-Hispanic according to ethnicity?

Response#2: Yes, having race and ethnicity within the same paragraph may be a little confusing. We have now described this better in the methods section. We have separated the two variables to illustrate that these questions were separate on the BRFSS and asked to everyone regardless of their race (see line 160).

Comment#3: Statistical Analysis: Correctly, the authors have first obtained a descriptive analysis of their population. Then they have used logistic regression due to the fact diabetic retinopathy is dichotomous (yes or no). I wonder, however, if a different approach could have been more satisfactory. For example, the authors could have tried to find clusters of conditions that are more frequently associated with diabetic retinopathy. In this case multiple component analysis, using diabetic retinopathy as a passive variable could have been more satisfactory. The authors should explain their choice more in detail.

Response#3: Thank you for this question. The definition of the outcome for our research question was based on the availability of questions related to eye complications due to diabetes included in the BRFSS survey. When analyzing the BRFSS for applicable questions regarding diabetes and diabetic retinopathy, we were only able to find the question “Has a doctor ever told you that diabetes has affected your eyes or that you had retinopathy?” This question included dichotomous ‘yes or no’ responses, hence our use of a logistic regression model. There was one additional question in the BRFSS that we noted, which stated “When was the last time you had an eye exam in which the pupils were dilated, making you temporarily sensitive to bright light?” However, we felt that this question was both vague and that we would not be able to relate it to diabetes and diabetic retinopathy. Finally, our aim in our study was to identify primarily independent SDOH related to DR as there is very little evidence available. The results of our study may be used to analyze clusters of conditions in a future study that may build upon the results revealed in our data.

Comment#4: Lines 183: “Most of the participants reported having health care coverage (n=13,985, 90.9%)”. On the basis of this observation it appears more reasonable not to consider this as a variable since almost all of the population of the study had health coverage. It could be interesting to know which factors are associated with DR in the small proportion of patient without a health insurance. Moreover “…a majority of participants reported living in an urban area (n=11,487, 86.9%)” (line 189). Are these the same who also have health coverage? If it is so it would be interesting to compare them with participants living in rural areas and without health coverage.

Response#4: Thank you for this very important comment. After running additional analyses to explore this, our team decided not to present the model stratified by insurance status, since after doing this, the standard errors (SE’s) were inflated/became too large.  For example, in the full adjusted model that included only participants with insurance coverage, the standard errors (SE) increased for the employment status variable. It was not possible to run the adjusted model for those participants who reported no health insurance coverage because only 57 participants reported DR and lived in a rural area, and several SE’s were inflated as a result. The cross tabulations between the urban/rural variable stratified by health insurance status (defined as Yes and No), both have p-values >0.05. Although for participants that reported no health insurance coverage, the prevalence of DR is higher in those who live in rural areas than those that live in urban areas (40% vs. 21.5%, p-value=0.123). We feel that the discussion of point estimates and 95% CI based on the adjusted analysis arrive at the same conclusion and are based on our main results.

Comment#5: Line 65 “…on how built environment effects DR…” Do the authors really mean that built environment produces DR, or may be they are saying that it is one of the factors that affects DR?

Response#5: Yes, we meant to say “affect” and not “effect.” Thank you for pointing out this mistake.  (See line 93)

Comment#6: Lines 195 “…Those ages 45-64” maybe should be typed “…Those aged 45-64”.

Response#6: Thank you for pointing this out. We changed “ages” to “aged.” (Line 226)

Comment#7: 3.2 heading should be placed between line 194 and line 195.

Response#7: Thank you for this comment, we have added a 3.2 heading, line 225.

Round 2

Reviewer 3 Report

Dear Authors,

I appreciated your efforts to improve the manuscript. I also appreciate your correctness and kindness in answering to my observations.
In particular, with the new title, the paper is now placed in a more correct perspective.
I hope that these useful research can continue to fill the gaps in the knowledge on socio-economic vulnerability as illness factor.
Merry Christmas and Happy New Year!